# Efficient and Workload-Aware LLM Serving via Runtime Layer Swapping and KV Cache Resizing

## Abstract

Efficiently serving large language models (LLMs) under dynamic and bursty workloads remains a key challenge for real-world deployment. Existing serving frameworks and static model compression techniques fail to adapt to workload fluctuations, leading to either service-level objective (SLO) violations under full-precision serving or persistent accuracy degradation with static quantization. We present *MorphServe*, a dynamic, workload-aware LLM serving framework based on *morphological adaptation*. MorphServe introduces two asynchronous, token-level runtime mechanisms: *quantized layer swapping*, which selectively replaces less impactful layers with quantized alternatives during high-load periods, and *pressure-aware KV cache resizing*, which dynamically adjusts KV cache capacity in response to memory pressure. These mechanisms enable state-preserving transitions with minimum runtime overhead and are fully compatible with modern scheduling and attention techniques. Extensive experiments on Vicuna and Llama family models with real-world workloads demonstrate that MorphServe reduces average SLO violations by 92.45% and improves the P95 TTFT latency by $2.2\times$–$3.9\times$ compared to full-precision serving, without compromising generation quality. These results establish MorphServe as a practical and elastic solution for LLM deployment in dynamic environments.

## 1   Introduction

The rise of large language models (LLMs) has made efficient and reliable serving a core challenge in modern AI infrastructure. Systems like vLLM [26] and TGI [23] optimize throughput via PagedAttention [26] and continuous batching [51, 44, 20], but assume fixed-precision execution and stable workloads. In contrast, real-world LLM workloads are dynamic and bursty [48, 2], with fluctuating request rates and context lengths. Even brief load spikes can cause memory exhaustion or queueing delays, leading to SLO violations—e.g., higher time-to-first-token (TTFT) and time-per-output-token (TOPT)—that degrade user experience and system throughput.

One naive solution is to statically over-provision GPU resources to accommodate worst-case traffic spikes. However, over-provisioning leads to substantial cost inefficiencies during underutilized periods [24, 15]. Moreover, edge deployments lack the flexibility for dynamic scaling altogether [4]. Thus, the inability to elastically match model resource usage to real-time demand results in either SLO violations under pressure, or significant resource waste during low-load intervals.

Model compression techniques, such as quantization [29, 13, 28, 39], pruning [33, 41, 17], or low-rank approximation [21, 49], offer an alternative approach by statically reducing the resource footprint of deployed LLMs. While these methods are effective in lowering memory and compute demands, they introduce irreversible accuracy degradation that persists even during periods of low load, when full-precision inference could be served without penalty. This results in a rigid, suboptimal quality–efficiency tradeoff that fails to align with workload variability. Key-value cache (KVC) compression [5, 54, 27] and eviction [31, 12] methods have been proposed to further reduce memory usage. However, these techniques often rely on fixed heuristics, cannot adapt to different workloads,

Submitted to 39th Conference on Neural Information Processing Systems (NeurIPS 2025). Do not distribute.

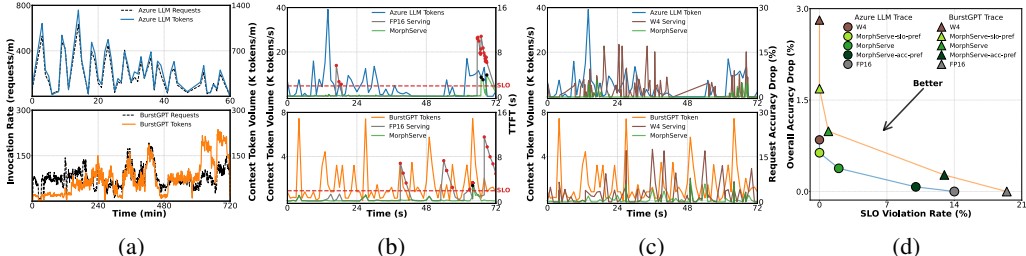

Figure 1: **Motivation for dynamic adaptation design in LLM serving.** (a) Real-world LLM workloads are highly dynamic and bursty in request and token volume. (b) Full-precision serving suffers TTFT spikes and SLO violations when workload exceeds the saturation point. (c) Statically quantized model causes constant accuracy degradation even during low-load periods when it is possible to serve full-precision models. (d) MorphServe dynamically adapts to resource pressure and consistently achieves an optimal balance between SLO compliance and accuracy.

lack compatibility with modern attention variants like Grouped Query Attention (GQA) [1, 7, 8] and Multi-Head Latent Attention (MLA) [34, 30], and remain inflexible to runtime serving conditions.

In this paper, we present MorphServe, a dynamic, workload-aware LLM serving framework based on morphological adaptation. MorphServe continuously monitors system load and morphs model components—transformer layers and KVC blocks—on the fly in response to real-time memory pressure. When resource usage surges, MorphServe reduces model footprint by replacing selected full-precision layers with lightweight quantized alternatives and expands KVC capacity by dynamically attaching additional memory blocks. These adaptations are reversed as pressure subsides, restoring full precision and reclaiming memory from KVC without interrupting inference.

MorphServe contributes the following: (1) A *runtime layer swapping* mechanism that enables *workload-aware mixed-precision serving*, allowing quantized and full-precision layers to coexist and be dynamically reconfigured based on runtime pressure without model flushing or architectural changes. (2) A *pressure-aware KVC resizing* mechanism that elastically adjusts KV cache capacity, supporting efficient batch prefilling and decoding under bursty traffic. (3) A tunable runtime policy that *navigates the accuracy–latency Pareto frontier*, balancing high-fidelity and low-latency objectives. (4) Full compatibility with existing KVC compression and eviction schemes, enabling further efficiency gains with minimal accuracy degradation.

To achieve this, MorphServe introduces two complementary morphing mechanisms, both designed to support asynchronous and compatible kernel executions with minimal overhead: LayerSwapper identify low-impact transformer layers by a sensitivity-based profiling, selectively and asynchronously replacing them with lower-precision alternatives at runtime. KVResizer adaptively adjusts KVC capacity under memory pressure and runs in parallel with decoding using separate CUDA streams, ensuring seamless execution.

Across extensive experiments on Llama 2 [46], Llama 3 [18], CodeLlama [38], and Vicuna [45] using four datasets [22, 53, 19, 11] under Azure LLM Inference [2] and BurstGPT [48] traces, MorphServe reduces average SLO violations by 92.45% and P95 TTFT latency by 2.2×–3.9× over full-precision serving, while preserving comparable accuracy. Compared to static quantization via AWQ [29], MorphServe reduces F1 and Rouge-L degradation by up to 88.85% and improves memory utilization by 29.29%. These results demonstrate MorphServe's ability to adapt to dynamic workloads while balancing performance and responsiveness.

## 2   Background and Motivation

**Real-world LLM workloads are highly bursty.** LLM serving systems face highly dynamic and bursty traffic patterns in real-world scenarios. As shown in Figure 1a, the production workloads of Microsoft Azure LLM services [2, 43] and BurstGPT [48] reveal rapid fluctuations in both the request arrival rates (i.e., request bursts) and the volumes of tokens. These fluctuations reflect the *non-stationary nature* of practical LLM inference workloads, which deviates from the traditional assumptions of most serving schemes [26, 51, 24].

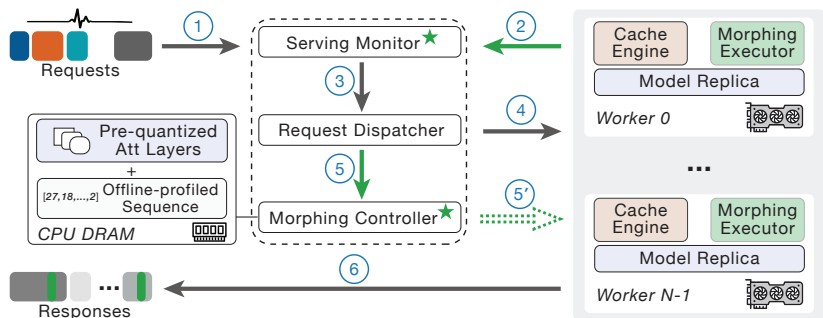

Figure 2: **MorphServe dynamic adaptation workflow.** Incoming requests (1) and real-time telemetry from workers (2) are aggregated by the Serving Monitor and sent to the Request Dispatcher (3). The Dispatcher routes new requests to workers (4) and forwards runtime metrics to the Morphing Controller (5), which detects resource pressure and issues adaptation commands to corresponding workers to reconfigure model layers and KVC (5). Each adaptation decision follows an offline-profiled layer swapping sequence. Quantized model variants at different precision levels are preloaded into CPU-pinned memory. Responses are returned to users (6), with only a small portion of tokens generated by mixed-precision layers (in green), enabling seamless adaptation.

**Request burst leads to long TTFT and SLO violation .** As system load increases, even small surges can cause sharp spikes in time-to-first-token (TTFT) latency. In this work, we set the TTFT SLO threshold to 2 seconds, consistent with prior work [50, 16, 36]. As shown in Figure 1b, full-precision serving quickly exceeds the SLO threshold once it reaches the **saturation point**—*defined as the load level at which available GPU memory becomes insufficient to schedule new requests for prefilling or to continue decoding for the ongoing batch.* At this point, incoming requests are forced to wait until memory is reclaimed, *incurring significant queueing latency with SLO violation.*

**Static quantization trades quality for efficiency irrespective of load.** To mitigate resource constraints, static quantization methods [29, 13, 28] have been widely adopted. However, these methods introduce persistent accuracy degradation across all conditions, regardless of whether the system is overloaded. As shown in Figure 1c, the INT4 quantized model with AWQ [29] consistently degrades accuracy—measured by F1 score following [28]—on the `GovReport` dataset from LongBench [3], even during periods when full-precision inference is feasible. This demonstrates that static quantization over-prioritizes efficiency, sacrificing model quality during low-load intervals.

**Workload-aware adaptation achieves optimal tradeoffs.** As shown in the Pareto analysis in Figure 1d, MorphServe achieves superior tradeoffs by aligning dynamic quantization with real-time workload demand. *A key insight is flexible, elastic mixed-precision LLM serving, where quantized and full-precision layers coexist and are dynamically reconfigured within a model in response to workload shifts.* This contrasts with static quantization and recent dynamic methods [6, 14, 37], which prioritize serving performance or hardware efficiency but overlook runtime workload variability. Most importantly, MorphServe enables smooth navigation along the efficiency—accuracy Pareto frontier—from uncompressed, high-accuracy models to highly quantized, efficient ones.

## 3 System Design

MorphServe is designed with three primary objectives: (1) Dynamic adaptation: Respond to real-time workload demands and GPU memory pressure by dynamically adjusting model layer configuration and KVC capacity on the fly during inference. (2) Accuracy preservation: Ensure no degradation under light or moderate load, and introduce only minimal, necessary, and fine-grained token-level accuracy loss to sustain serving performance beyond the saturation point. (3) Low overhead: Minimize the performance impact of dynamic adaptation by leveraging asynchronous execution and overlapping.

### 3.1 Architecture and Workflow

**System Architecture.** As illustrated in Figure 2, MorphServe consists of three core components—*Serving Monitor*, *Morphing Controller*, and *Morphing Actuator*, which together form a feedback-driven control loop for dynamic adaptation.

- *Serving Monitor* collects runtime metrics from all workers, including GPU memory utilization, request queue depth, throughput, and token-level latency (TTFT and TPOT). These metrics are smoothed over short time windows to identify workload shifts and early signs of system saturation.
- *Morphing Controller* serves as the global GPU memory manager. When monitored metrics exceed user-predefined thresholds (e.g., KVC memory usage > 85%, queueing delay > 100 ms), it decides whether to trigger *selective layer swapping* (Section 3.3) and *elastic KVC resizing* (Section 3.4), and dispatches corresponding instructions to the target workers.
- *Morphing Actuator* resides on each worker and executes adaptation commands locally. It dynamically reconfigures the model using LayerSwapper (Section 3.3), which switches a selective set of layers between full-precision and pre-quantized layers, or between different quantization levels (e.g., from INT8 to INT4) to reduce resource usage and improve inference latency under pressure. In addition, it applies KVResizer (Section 3.4) to adjust KVC memory allocation by elastically expanding or shrinking the number of KVC blocks as needed. All adaptations are asynchronous overlapping communication and computation [42] with preallocated memory buffers to seamlessly overlap with ongoing inference.

MorphServe's adaptive and versatile architecture enables efficient and timely operation across diverse and bursty workloads, ensuring serving quality under pressure while avoiding unnecessary degradation during underloaded periods.

**Token-level Workload Adaptation.** Unlike existing model and KVC compression schemes, which affect the entire request [35, 29, 13, 28, 52], MorphServe enables *fine-grained, token-level* workload adaptation. During a single request's decoding phase, MorphServe may temporarily replace a subset of layers. For example, switching 2 layers from full-precision to INT4 when saturation is detected (examples shown in Section 3.3). This allows early tokens to be generated at full precision, while only later tokens experience minimal accuracy degradation. Once the pressure subsides, the affected layers are restored to full precision, enabling continued decoding at the original accuracy. As a result, accuracy degradation is confined to a small portion of tokens, even within a single request.

**State-Preserving Morphing During Inference.** A key feature of MorphServe's serving workflow is its ability to seamlessly adapt model layer precision and elastically resize KVC capacity on-the-fly during request execution, *without model flushing or re-prefilling*. When system pressure triggers adaptation, the Morphing Controller can selectively swap model layers without disrupting the attention state or decoding progress, avoiding expensive *serving pauses* and *recomputation*. This design allows MorphServe to intervene mid-inference at the token level, preserving continuity in generation and enabling real-time adaptation with minimal runtime interference.

### 3.2 Offline Profiling for Layer Swapping Sequence

To identify a layer swapping sequence that minimizes accuracy impact during runtime, MorphServe performs *offline profiling* to construct a prioritized swapping order based on sensitivity analysis. In this subsection, we describe how MorphServe profiles and ranks layers to establish this sequence with a focus on accuracy and robustness.

**Problem Statement.** The objective is to minimize cumulative accuracy degradation over the time interval during which one or more layers are quantized.

Let $f(x_t)$ denote the full-precision model output at time $t$, and $f^{(Q_t)}(x_t)$ the output when a subset of layers $Q_t$ are quantized at that time. The cumulative degradation over the interval $[t_1, t_n]$ can be formulated as:

$$\min_{\{n_k\}} \sum_{t=t_1}^{t_n} \Delta(f(x_t), f^{(Q_t)}(x_t)) \tag{1}$$

This problem has a sequential and state-dependent structure: each swapping decision impacts downstream accuracy until the corresponding layer is restored to full precision. Selecting which layers to replace introduces combinatorial complexity, making exact optimization intractable. To address this, MorphServe performs offline profiling using hybrid sensitivity metrics to evaluate the accuracy impact of each layer. The resulting sequence provides a prioritized order of layers that can be replaced with minimal expected accuracy degradation. We now describe the sensitivity metrics and the greedy policy used to construct this sequence.

**Sensitivity Analysis for Layer Swapping.** To construct the swapping sequence, MorphServe estimates the sensitivity of each decoder layer using cosine similarity-based local and global metrics

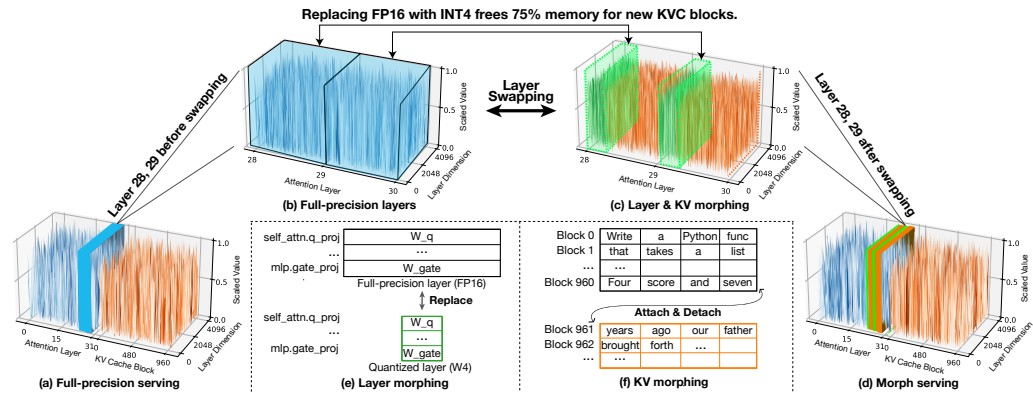

Figure 3: **Synergy of dynamic layer swapping and elastic KVC resizing.** Figure (a)–(d) illustrate the model state morphing process: starting from full-precision serving (a), selected layers (b) are replaced with quantized versions (c) without disrupting the inference computation. This process leads to mixed-precision layer serving (d). Figure (e) shows the detailed decoder layer swapping mechanism. Figure (f) demonstrates KVC block management under `KVResizer`, where newly vacant memory blocks are dynamically reallocated to KVC or deallocated from KVC based on real-time workload shifts. `KVResizer` reduces the request preemption rate for decoding and incoming request queueing time for prefilling.

that capture its impact on overall model accuracy. These sensitivity scores are used to rank layers, providing a prioritized order that approximates the optimal swapping strategy.

- *Layer Transformation Sensitivity (LTS)* measures the direct change between a layer's input and output:

$$\text{LTS}_p = \cos\left(h_p(x), x_p\right) \tag{2}$$

Where $x_p$ is the input and $h_p(x)$ is the output of layer $p$. Lower similarity indicates stronger transformations and higher potential sensitivity to layer swapping.

- *Layer Replacement Sensitivity (LRS)* quantifies the output distortion caused by replacing the original layer with its quantized version:

$$\text{LRS}_p = \cos\left(h_p(x), h_p^Q(x)\right) \tag{3}$$

where $h_p^Q(x)$ is the output of layer $p$ with quantized weights. Lower similarity implies greater deviation due to replacement.

- *Model Degradation Sensitivity (MDS)* measures the model-level accuracy impact from replacing a layer $p$ given the current set of quantized layers $Q$:

$$\text{MDS}_p^{(Q)} = \cos\left(f^{(Q)}(x), f^{(Q\cup\{p\})}(x)\right) \tag{4}$$

where $f^{(Q)}(x)$ is the model output with layers $Q$ replaced. This state-aware metric captures the incremental global degradation introduced by swapping layer $p$ in the current context.

We combine these metrics into a unified **Layer Importance Score (LIS)**:

$$\text{LIS}_p = \alpha_1 \cdot \text{LTS}_p + \alpha_2 \cdot \text{LRS}_p + \beta \cdot \text{MDS}_p^{(Q)} \tag{5}$$

In this formulation, $\text{LTS}_p$ and $\text{LRS}_p$ are *local sensitivity metrics* that evaluate the behavior of the layer $p$ in isolation, while $\text{MDS}_p^{(Q)}$ is a *global metric* that measures the model-level degradation when replacing $p$, given the current replaced layer set $\mathcal{Q}$. For a given model, the LIS for each layer is computed *offline during profiling*, and the resulting sequence is stored and used directly at runtime. This design avoids any runtime recomputation or decision-making overhead. Full details on the scoring, hyperparameter tuning, and selection algorithms are provided in the Appendix.

### 3.3 **LayerSwapper: Runtime Layer Swapping**

To enable efficient and non-disruptive layer replacement during inference, `MorphServe` leverages the precomputed layer swapping sequence from offline profiling to guide the dynamic runtime adaptation mechanism. This mechanism consists of two key components: (1) model preloading with kernel

precompilation, which ensures that both full-precision and quantized versions of layers are memory-resident and ready for execution; and (2) asynchronous layer swapping, which allows selected layers to be swapped between CPU and GPU memory on-the-fly without blocking inference.

- *Model Preloading and Kernel Precompilation.* Prior to serving, all decoder layer variants (e.g., FP16, INT8, INT4, and INT3) are preloaded into a contiguous, pinned CPU memory region, while the full-precision model replica is loaded into a preallocated contiguous GPU memory, as shown in the Figure 2. MorphServe tracks the memory addresses of all layer variants, enabling efficient direct memory copies for layer swapping. To avoid runtime latency, inference kernels corresponding to precision levels are precompiled in advance. We also implement kernel fusion to optimize performance, while the rest of the serving pipeline reuses state-of-the-art techniques—such as PagedAttention [26] and FLASHATTENTION [10, 9]—to ensure compatibility and efficiency.

- *Asynchronous In-place Layer Swapping.* At runtime, MorphServe performs in-place layer swapping using asynchronous CUDA streams to avoid interference with ongoing decoding. As illustrated in Figure 3, when layers 28 and 29 are selected for replacement, the swapping process is launched asynchronously while earlier layers (e.g., 0–27) continue computation without interruption. Full-precision layers are safely discarded from GPU memory since their backup copies reside in pinned CPU memory, and quantized variants are copied into the same memory addresses to avoid pointer remapping. Due to the relatively compact size of each decoder layer (e.g., 0.4 GB for FP16 and 0.1 GB for INT4 in Llama 2 7B), the PCIe transfer latency is minimal - approximately 4 ms for INT4 and 16 ms for FP16 for Llama2 7B on PCIe Gen4 with up to 26-28 GB/s bandwidth. In practice, the complete layer swapping process for a INT4 variant—including memory transfer and reconstruction—takes approximately 6 ms and is fully overlapped with decoding, resulting in negligible TPOT overhead. Additional performance breakdowns are provided in Section 4.

### 3.4 KVResizer: Elastic KVC Resizing

To support bursty workloads and fluctuating memory demands, MorphServe integrates dynamic layer swapping with KVResizer, a mechanism for elastic resizing of key-value cache (KVC) blocks. This section addresses two key questions: (1) how KVResizer dynamically allocates and releases KVC blocks in response to runtime memory pressure, and (2) how it collaborates with layer swapping to maintain serving efficiency under peak load.

KVResizer is triggered when the Serving Monitor detects insufficient GPU memory to allocate KVC blocks for incoming request *prefilling* or ongoing *decoding*. To free memory, MorphServe initiates layer swapping, replacing selected full-precision layers with quantized variants. This reduces the model's memory footprint—e.g., replacing an FP16 layer with INT4 can save up to *75%* memory, as shown in Figure 3—enabling allocation of new KVC blocks.

KVResizer extends PagedAttention [26] with kernel-level support for *on-demand KVC block allocation/deallocation*, implemented through memory mapping without requiring kernel recompilation. All resizing operations are executed asynchronously using separate CUDA streams to avoid interference with ongoing decoding.

Unlike static preallocation strategies (e.g., in vLLM [26]), KVResizer adjusts KVC capacity dynamically based on real-time memory availability. Once the pressure subsides, both temporary KVC blocks and quantized layers are released and restored to their full-precision state, ensuring memory reuse and accuracy recovery.

As a result, KVResizer enhances system efficiency across both the prefilling and decoding phases under high-load conditions.

- *Reducing queueing Time and TTFT During Prefilling.* Under static scheduling, incoming requests may queue indefinitely when no GPU memory is available for KV allocation. Since FIFO schedulers typically release memory only after a request finishes decoding, long queueing delays directly translate into TTFT violations. In MorphServe, KVResizer is triggered when the queue length or wait time exceeds a threshold, proactively attaching new KV blocks to admit pending requests. This significantly reduces queueing time and improves TTFT under bursty traffic.

- *Reducing Preemption and Improving TPOT During Decoding.* In the decoding phase, requests are preempted if no KV blocks are available, forcing swaps to host memory or full recomputation, both of which introduce delays and degrade TPOT and end-to-end latency. By dynamically attaching

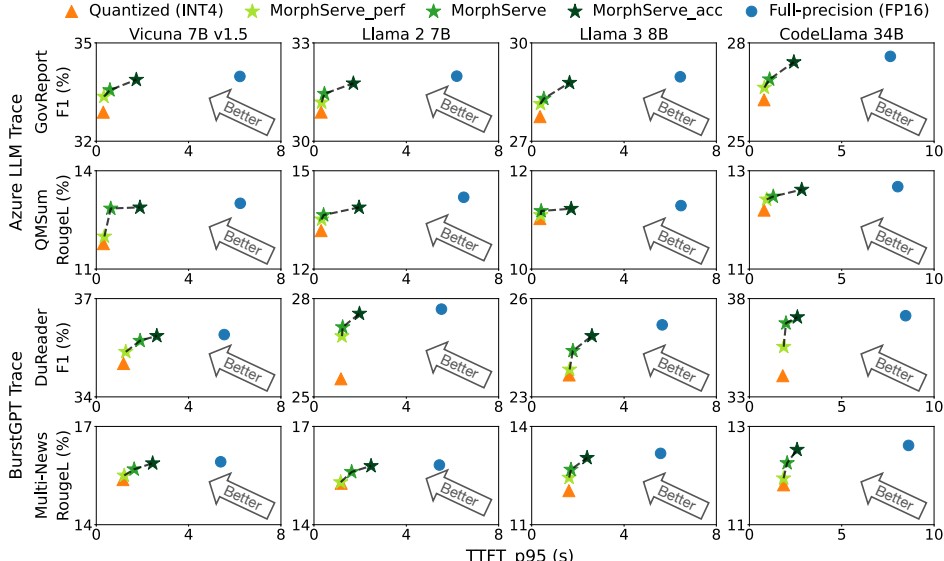

Figure 4: **MorphServe provides the best latency–accuracy tradeoff across four models and two traces, with four datasets.** MorphServe in accuracy mode (dark green) reduces P95 TTFT by 2.2×–3.9× compared to full-precision serving while maintaining comparable generation quality. In performance mode (light green), MorphServe consistently outperforms INT4 quantized models in output quality with no additional latency overhead.

KV blocks at runtime, MorphServe reduces preemption events and maintains decoding continuity, leading to better overall system responsiveness.

Together, these improvements enable MorphServe to utilize GPU memory more efficiently across load conditions, mitigate bottlenecks under saturation, and achieve a balanced trade-off between accuracy and responsiveness in volatile serving scenarios.

## 4 Experiment

**Evaluation Setup**. We evaluate MorphServe across a diverse range of LLM architectures, workload traces, and tasks. We consider four representative models: Vicuna 7B v1.5 [32], Llama 2 7B [46], Llama 3 8B [18], and CodeLlama 34B [38], spanning multiple scales and attention types—including Multi-Head Attention (MHA) [47] and Grouped-Query Attention (GQA) [1]. We test two real-world LLM inference workload traces: the BurstGPT trace [48] and the Azure LLM Inference trace [43, 2]. We report results from a representative 72-second trace snippet (Figure 1) for both workloads, though MorphServe is effective across the full traces. The request arrival rates of each trace is downscaled by 1.75× and 4.75× to fit our hardware environment. To evaluate generation quality, we use four public datasets: GovReport [22] and Multi-News [11] (long-form summarization), QMSum [53] (query-based summarization), and DuReader [19] (reading comprehension). For each test, we align workload timestamps with context passages from the datasets. Prompt and response lengths are set to *512* and *256* tokens for Vicuna 7B v1.5 and Llama 2 7B, and to *1024* and *512* tokens for Llama 3 8B and CodeLlama 34B. We report F1 and Rouge-L scores to assess generation quality. End-to-end experiments for Vicuna 7B v1.5, LLaMA 2 7B, and LLaMA 3 8B are conducted on an NVIDIA L4 GPU with 24 GB HBM and 256 GB of CPU DRAM, while CodeLLaMA 34B is evaluated on an A100 server with 80 GB HBM and 2 TB of CPU DRAM.

**Implementation.** MorphServe is implemented on top of SwiftLLM [25, 40], a lightweight and modular LLM inference framework that reproduces vLLM performance with simplified components. We added approximately 2,200 lines of Python and 500 lines of C++/CUDA to support MorphServe's optimized KVC management and attention kernel extensions, which enable efficient layer swapping and KVC resizing at runtime.

**Baselines.** We include the full-precision (FP16) model as an upper-bound reference and an INT4 quantized model as a static compression baseline. For quantization, we adopt AWQ [29] due to its efficient inference kernel support; however, the setup is compatible with any post-training quantization

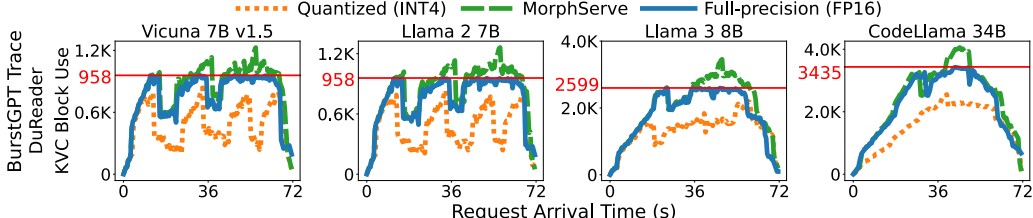

Figure 5: **MorphServe dynamically adapts KVC block capacity to workload fluctuations.** The red line indicates the KV cache capacity under full-precision serving. MorphServe (green) elastically attaches new KV blocks during peak loads, pushing the saturation boundary and preventing request preemption or KVC swapping in the full-precision baseline (blue). Static quantization (orange) underutilizes memory due to its fixed configuration, even when resource headroom is available.

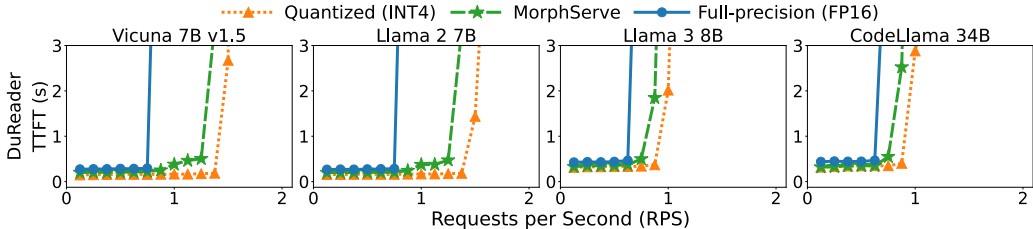

Figure 6: MorphServe delays saturation and achieves up to $1.83\times$ throughput than full-precision serving under increasing request rates.

method and can be replaced accordingly. To evaluate the flexibility of MorphServe, we configure it in two runtime modes: In *accuracy mode*, MorphServe prioritizes output quality by raising the threshold for triggering layer swapping and limiting the number of quantized layer replacements, thereby minimizing accuracy degradation. In *performance mode*, MorphServe enables more aggressive layer swapping to improve throughput and reduce latency under memory pressure. All baselines and MorphServe configurations are evaluated on the same serving engine to ensure fair comparison.

## 4.1 Main Results

**TTFT and Accuracy.** As shown in Figure 4, MorphServe significantly reduces P95 TTFT latency while preserving output quality in all model-trace-dataset configurations. Compared to full-precision baselines, MorphServe reduces the P95 TTFT by $2.9\times$–$15.7\times$ ($2.2\times$–$3.9\times$ in accuracy mode and $3.4\times$–$19.5\times$ in performance mode) while maintaining quality within $0.51\%$–$3.82\%$ degradation on F1 or Rouge-L scores, as low as $0.11\%$–$2.18\%$ in accuracy mode. In contrast, static quantization exhibits $2.34\%$-$9.47\%$ degradation compared to full-precision inference, due to suffering from persistent quality loss across the entire serving lifetime. In particular, MorphServe excels in long-context datasets such as GovReport, leveraging LayerSwapper and KVResizer to optimize memory and computing efficiency. MorphServe with different configurations (green stars) visualizes the ability to navigate the latency-accuracy Pareto frontier, offering the best balance of performance and quality based on real-time workload shifts.

**Workload Adaptation and Saturation Resilience.** As shown in Figure 5, MorphServe adaptively manages KVC block capacity in response to fluctuating load. In the full-precision baseline, KVC usage saturates the static capacity limit during peak periods, resulting in elevated queueing delays, request preemption, and frequent KVC swapping, which can lead to SLO violations. Static quantization, while reducing the memory footprint, degrades model accuracy and underutilizes GPU memory, even during low-load periods. MorphServe attaches new blocks during bursty traffic and releases them as load subsides, enabled by the synergistic LayerSwapper and KVResizer mechanism. MorphServe improves overall KVC memory utilization and output accuracy by $29.29\%$ and $3.58\%$, respectively, compared to static quantization. The adaptation allows MorphServe to expand KVC usage by up to $32.97\%$ beyond the full-precision limit when needed, and reduce the queueing delay by up to $3.8\times$. MorphServe also mitigates request preemption and KVC swapping under saturation conditions. This enhances system responsiveness and improves token-level efficiency, contributing to reduced TPOT and end-to-end request latency.

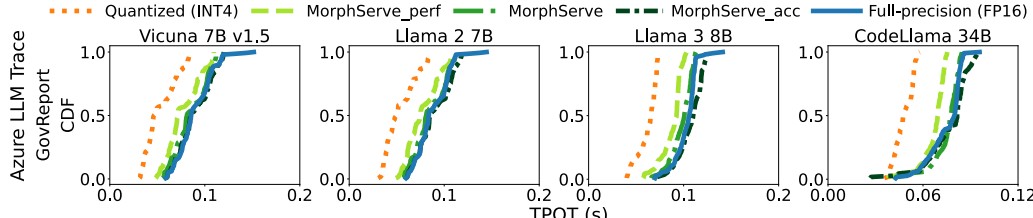

Figure 7: **MorphServe incurs negligible runtime overhead while improving tail TPOT latency.** MorphServe (green) achieves comparable average TPOT latency to the full-precision baseline (blue), while reducing P99 latency by up to $1.23\times$. Performance mode (light green) improves the average TPOT by up to $1.17\times$ through aggressive layer morphing.

**Throughput.** In Figure 6, we compare MorphServe with baselines on DuReader under varying request rates. All configurations maintain low TTFT at low RPS, but as load increases, full-precision inference encounters the threshold, where TTFT spikes abruptly due to memory exhaustion and queueing delays. In contrast, MorphServe consistently pushes back this saturation point, achieving $1.6\times$–$1.83\times$ higher throughput than full-precision serving across all evaluated models.

**CPU Memory Overhead.** Compared to full-precision, MorphServe introduces modest additional host memory usage by maintaining a mixed set of full-precision and quantized variants of transformer layers. Fortunately, the overhead is bounded: quantized weights (e.g., W8, W4, W3) are significantly smaller than their full-precision counterparts, and the combined memory footprint typically does not exceed $2\times$ the original model size. Moreover, multi-GPU deployment of large models or model replicas hosted on multiple GPUs within the same node can share a single copy of quantized, CPU-memory-residential model weights across GPUs, eliminating redundant CPU memory consumption. In our experiments on an NVIDIA A100 $8\times$ 80GB server with 2 TB of host memory, the total memory footprint, including both swapped-out full-precision layers and INT4 quantized variants of CodeLlama 34B, accounted for only $4.42\%$ of available host memory, introducing negligible memory bandwidth and capacity overhead. These results confirm that MorphServe's host memory footprint is practical and sustainable for both cloud-scale and high-end edge deployments.

**Runtime Performance Overhead.** Figure 7 presents the cumulative distribution (CDF) of time-per-output-token (TPOT) across two datasets and four models under the Azure LLM trace. MorphServe delivers average TPOT comparable to full-precision serving while improving P95 and P99 TPOT tail latency by up to $1.06\times$ and $1.23\times$, respectively. These gains are achieved by eliminating request preemption stalls and avoiding KVC swapping or recomputation—two primary sources of long-tail delays. The performance mode of MorphServe reduces average TPOT by $1.11\times$-$1.17\times$, while the accuracy mode introduces overhead of up to $1.06\times$ as it preserves more full-precision layers and applies stricter thresholds for layer swapping. In accuracy mode, this conservative strategy increases memory usage and may lead to occasional queueing delays under load. The TPOT gain from MorphServe is due to faster inference on quantized layers, and the highly efficient kernels on layer swapping (e.g., $\sim$6 ms for a Llama 2 7B INT4 attention layer). These results confirm that MorphServe introduces negligible runtime overhead while effectively reducing tail latency.

MorphServe supports an optional offline calibration step to pre-compute layer sensitivity scores for more accurate morphing decisions. While this process improves accuracy-latency tradeoffs, it is not required for MorphServe to function. For a model, the calibration is a once-for-all process. Given its offline nature and minimal duration, the overhead is negligible and acceptable in practice. Details of the calibration procedure and associated cost are provided in the Appendix.

## 5 Conclusion

This paper presents MorphServe, a novel workload-aware LLM serving framework based on morphological adaptation. MorphServe dynamically adjusts model precision through LayerSwapper and KVC memory capacity through KVResizer, in a coordinated manner based on real-time resource usage. MorphServe maintains high-quality inference under normal conditions and adapts gracefully during overload periods. Our design achieves substantial improvements in SLO compliance rates, memory efficiency, and serving robustness, while incurring minimal quality loss and runtime overhead.

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
