# OpenReview forum: "Efficient and Workload-Aware LLM Serving via Runtime Layer Swapping and KV Cache Resizing"
_ICLR.cc/2026/Conference — ICLR 2026 Conference Withdrawn Submission_

### Official Review · Reviewer_qMi5 · 2025-10-27

**Soundness:** 2
**Presentation:** 2
**Contribution:** 2
**Rating:** 2
**Confidence:** 4

**Summary:**

This paper introduces MorphServe, a dynamic LLM serving framework that adapts to fluctuating workloads via runtime layer swapping and elastic KV cache resizing. It employs morphological adaptation to selectively replace less critical model layers with quantized versions and adjust KV cache capacity based on real-time memory pressure. This approach preserves high accuracy during low load and only introduces minimal, targeted precision loss to maintain performance during high load, without interrupting ongoing requests. Extensive experiments show MorphServe significantly reduces latency and SLO violations while maintaining model quality, outperforming both full-precision and static quantization baselines.

**Strengths:**

* S1) The proposed method enable dynamic trade-off between model quality and serving efficiency.
* S2) The proposed method achieves significant latency reduction and SLO improvement with negligible accuracy loss

**Weaknesses:**

* W1) The authors incorrectly claim in the introduction that key-value cache compression and eviction methods are incompatible with Grouped Query Attention. To my knowledge, nearly all recent KV cache eviction techniques have been successfully applied to GQA-based models, including Llama 3 and Qwen 3.

* W2) The models used in the experiments are somewhat outdated.

* W3) MorphServe heavily relies on offline profiling to determine layer sensitivity and swapping sequences. This process requires preprocessing for each specific model on particular datasets to compute Layer Importance Scores (LIS).

* W4) The authors used the NeurIPS 2025 template instead of the ICLR template.

**Questions:**

please refer to Weaknesses.

---

### Official Review · Reviewer_CENS · 2025-10-30

**Soundness:** 2
**Presentation:** 2
**Contribution:** 2
**Rating:** 2
**Confidence:** 4

**Summary:**

Desk reject, the author uses the wrong template

**Strengths:**

Desk reject, the author uses the wrong template

**Weaknesses:**

Desk reject, the author uses the wrong template

**Questions:**

Desk reject, the author uses the wrong template

---

### Official Review · Reviewer_C6rS · 2025-10-30

**Soundness:** 3
**Presentation:** 4
**Contribution:** 3
**Rating:** 6
**Confidence:** 5

**Summary:**

This paper adaptively change the model weight precision during inference time, which allow more requests to start generation during workload peak and thus significantly reducing tail TTFT.

**Strengths:**

* Solid implementation with sufficient low-level details
* Interesting idea of changing the precision of model weights to fit in more requests

**Weaknesses:**

* Does not work with already-quantized LLMs.
* Evaluation coverage can be wider
* CPU memory overhead
* Slightly hurts TPOT

**Questions:**

I appreciate the core ideas, but mainly concerned about the compatibility of this work with existing LLM serving optimization techniques. For example:

* Does your approach support CUDA graph?
* How does your approach support different parallelism? Like tensor parallel, pipeline parallel, pd disaggregation and wide ep?
* The evaluation is on old LLMs --- what about latest LLMs, like MoE models? Will the improvement be larger or smaller?
* Your approach will likely slow down CPU KV cache offloading. If that is the case, a small experiment on how much the slowdown would be will be helpful.
* How does your approach compared to other layer-wise kv cache management approaches like this paper https://arxiv.org/html/2410.00428v1 ? I see that you are citing this work, but the rationale of why your work is better is missing. I understand that your approach is working on an orthogonal dimension, but your paper and this paper is targeting the same problem.

---

### Official Review · Reviewer_Wj4H · 2025-10-31

**Soundness:** 2
**Presentation:** 1
**Contribution:** 2
**Rating:** 2
**Confidence:** 5

**Summary:**

The paper proposes a workload-aware LLM serving framework that dynamically switches between full-precision (FP) and quantized model layers to manage memory pressure. The core idea is that when a server is overloaded, the system swaps a subset of FP layers with their quantized counterparts. This action has two benefits: 1) newly generated KV-cache entries are smaller, and 2) the system's "KVResizer" component can allegedly resize the _existing_ KV-cache blocks corresponding to the swapped layers. This dynamic memory saving is intended to increase request throughput and reduce time-to-first-token (TTFT) by accommodating more concurrent requests, though with a marginal, controlled degradation in model accuracy. The evaluation shows improved TTFT and F1 scores against selected baselines.

**Strengths:**

- The central concept of using dynamic, partial quantization to actively manage the KV-cache memory bottleneck is novel. If the proposed resizing of the existing KV cache can be achieved with minimal overhead, it represents a promising direction for handling memory-intensive serving workloads.
- The use of cosine similarity to define Layer Transformation Sensitivity (LTS) and Layer Replacement Sensitivity (LRS) is intuitive. It provides a clear, model-aware heuristic for identifying which layers are most (and least) sensitive to quantization, guiding the layer-swapping policy.

**Weaknesses:**

- Overly naive and brute force solution
- Missing details
- Missing Appendix and formatting issues

Details:
- The method for determining the optimal set of layers to quantize (MDS score) appears to be a brute-force search. The paper defines the score for a layer $p$ given a set of already quantized layers $Q$, but it never explains how this set $Q$ is chosen. Without a clear heuristic or optimization, the algorithm would need to explore $O(2^n)$ combinations of layers to find the optimal subset, which is computationally infeasible for modern LLMs.
- The "KVResizer" is a central component of the proposed system, yet its implementation is treated as a black box. A key claim is that it can "shrink" the KV-cache blocks of _existing_, ongoing requests. How is this resizing performed on an active cache block while a decoding process is concurrently accessing it? This operation seems non-trivial, likely requiring significant, complex CUDA kernel development to manage memory and synchronization without stalling the GPU. The paper provides no details, overhead analysis, or evidence to support the feasibility of this core mechanism.
- The paper is not in the ICLR format. I reviewed the paper regardless and my review does not depend on this. However, I want to point it out so AC or SAC can make an informed decision. More critically, it repeatedly refers to an appendix for “key details” (including, presumably, information on the layer swapping and resizing mechanisms), but no appendix or supplementary material is provided. This omission makes it impossible to fully evaluate the paper's technical contribution. An LLM usage statement is also missing.
- The mathematical notation is confusing. For example, $Q$ is used in Equation 3 to represent a quantized layer ($h_p^Q$​) but is then redefined in Equation 4 as a subset of layers being swapped.

**Questions:**

1. The paper claims (Line 208) that layer loading is overlapped with decoding. How is this possible? Doesn't the decoding process require access to the very layer weights that are being swapped in or out?
2. The system loads all layer variants into CPU memory before serving. What is the total CPU memory footprint required for a typical model, and is this a practical assumption for a production serving node?
3. The paper states that experiments are run on a "72-second trace snippet". This duration seems exceptionally short for evaluating a dynamic serving system. Were all experiments limited to this 72-second window? This is insufficient to evaluate system stability, long-term behavior (e.g., memory fragmentation), or the overhead of repeated swapping.

**Details Of Ethics Concerns:**

The paper is not in the ICLR format and is missing appendix and LLM usage statement.

---

### Note · Authors · 2025-11-12

I have read and agree with the venue's withdrawal policy on behalf of myself and my co-authors.